# Detection of *Salmonella* Typhimurium in Romaine Lettuce Using a Surface Plasmon Resonance Biosensor

**DOI:** 10.3390/bios9030094

**Published:** 2019-07-28

**Authors:** Devendra Bhandari, Fur-Chi Chen, Roger C. Bridgman

**Affiliations:** 1Department of Agricultural and Environmental Sciences, Tennessee State University, Nashville, TN 37209, USA; 2Department of Human Sciences, Tennessee State University, Nashville, TN 37209, USA; 3Hybridoma Facility, Auburn University, Auburn, AL 36830, USA

**Keywords:** biosensor, surface plasmon resonance, *Salmonella* Typhimurium, flagellin, monoclonal antibodies

## Abstract

Leafy vegetables have been associated with high-profile outbreaks causing severe illnesses. Timely and accurate identification of potential contamination is essential to ensure food safety. A surface plasmon resonance (SPR) assay has been developed for the detection of *Salmonella* Typhimurium in leafy vegetables. The assay utilizes a pair of well characterized monoclonal antibodies specific to the flagellin of *S*. Typhimurium. Samples of romaine lettuce contaminated with *S.* Typhimurium at different levels (between 0.9 and 5.9 log cfu/g) were pre-enriched in buffered peptone water. Three SPR assay formats, direct assay, sequential two-step sandwich assay, and pre-incubation one-step sandwich assay were evaluated. All three assay formats detect well even at a low level of contamination (0.9 log cfu/g). The SPR assay showed a high specificity for the detection of *S.* Typhimurium in the presence of other commensal bacteria in the romaine lettuce samples. The results also suggested that further purification of flagellin from the sample preparation using immunomagnetic separation did not improve the detection sensitivity of the SPR assay. The functional protocol developed in this study can be readily used for the detection of *S.* Typhimurium in leafy vegetables with high sensitivity and specificity.

## 1. Introduction

Non-typhoidal *Salmonella* is one of the leading causes of foodborne illnesses in the world [1]. In the United States, *Salmonella* is responsible for 1,027,561 cases of illnesses, 19,336 hospitalizations, and 378 deaths annually [2]; which result in a direct and indirect economic cost of $3.3 billion [3]. *Salmonella enterica* serovar Typhimurium (*S.* Typhimurium) is the second most common serotype (after *Salmonella enterica* serovar Enteritidis) that causes foodborne illnesses [4]. These public health issues and enormous economic costs associated with illnesses mandate the need for rapid, sensitive, and specific *S.* Typhimurium detection methods.

The culture based detection methods require intensive labor and take 3–4 days for the preliminary identification and 5–7 days for the confirmation [5]. Use of chromogenic and fluorogenic growth media in the culture based methods had reduced the detection time by days, but this was not fast enough to respond to disease outbreaks and product recalls [6]. Immunological methods like enzyme-linked immunosorbent assay (ELISA) are faster than the cultural methods [7], and are commonly used in the detection of *S.* Typhimurium [8,9,10,11]. However, lower sensitivity, requirement of pre-enrichment step and need for sample pre-treatment in ELISA still left the avenues to develop faster and more sensitive detection methods. Polymerase chain reaction (PCR) based molecular methods are faster and more sensitive than ELISA, and have been extensively studied for the detection of *S.* Typhimurium in foods [12,13,14,15,16]. Major drawbacks of the PCR detection methods are difficulty in automation, requirement of sample pre-enrichment [17,18] and false-negative results due to PCR inhibitors in the samples [19,20,21].

Recently the surface plasmon resonance (SPR) biosensor has gained attentions in foodborne pathogen detection because SPR assays are fast, label-free, and allow real-time monitoring of the biomolecular interactions with high sensitivity and specificity [22]. There have been limited studies on the detection of *S.* Typhimurium in food matrixes using SPR biosensors [23,24,25,26,27]; yet there is no report of the SPR assay for the detection of *S.* Typhimurium in leafy vegetables such as romaine lettuce. Most of these studies have used polyclonal antibodies to directly detect bacterial cells from food matrixes. However, polyclonal antibodies (mixture of various antibodies) have the tendency to bind with non-target antigens. This cross reactivity (nonspecific binding) is a major concern in many detection methods. In addition, use of bulk sized bacteria (0.7-1.5 × 2.0-5.0 µm) may hinder the binding of antibody with antigen in the SPR assay and result in the reduction of detection sensitivity.

The problem of nonspecific binding of polyclonal antibodies associated with *S.* Typhimurium in the SPR assay can be circumvented by using monoclonal antibodies specific to the targeted antigens. Monoclonal antibodies specific to flagellin of *S.* Typhimurium have been produced [28] and characterized in terms of their binding kinetics and epitope maps, as described in our previous work [29]. Use of the well characterized monoclonal antibodies either in direct or sandwich assays will render the SPR assay more specific and sensitive. The purpose of this study is to develop a sensitive and specific SPR assay for the detection of *S.* Typhimurium in romaine lettuce. Different assay formats and their sensitivity and specificity were evaluated and a functional protocol for the routine application was developed.

## 2. Materials and Methods

### 2.1. Materials and Instrument

*S.* Typhimurium (ATCC 13311) was acquired from American Type Culture Collection (Manassas, VA, USA) and stored at −80 °C before use. *Enterobacter cloacae*, *Pseudomonas fluorescens*, *Pseudomonas aeruginosa*, *Aeromonas salmonicida*, *Photobacterium damselae*, *Serratia* spp., and *Brucella* spp. were isolated in our laboratory from the romaine lettuce purchased from a local grocery store.

Tryptic soy agar (TSA), xylose-lysine-tergitol 4 (XLT-4) agar, and buffered peptone water (BPW) were supplied by Thermo Fisher Scientific Inc. (Lenexa, KS, USA). Difco plate count agar (PCA) and BBL lactose broth (LB) were purchased from Becton, Dickinson and company (Sparks, MD, USA). Bovine serum albumin (BSA), 10X phosphate buffered saline (PBS), and TWEEN 20 were obtained from Fisher Scientific (Fair Lawn, NJ, USA). PBST (1X PBS with 0.05% TWEEN 20) was prepared in our laboratory and used as working buffer and SPR running buffer. API-20E identification kits were purchased from bioMérieux, Inc. (Durham, NC, USA).

*N*-(3-Dimethylaminopropyl)-*N*′-ethylcarbodiimide hydrochloride (EDC), *N*-Hydroxysuccinimide (NHS), ethanolamine hydrochloride, sodium acetate, and glycine were acquired from Sigma-Aldrich Inc (St. Louis, MO, USA). Deionized water was purified with a Millipore purification system (Simplicity Water Purification System) and then degassed with a vacuum chamber. All solutions were prepared in deionized degassed water.

Hula mixer and Dynal magnet and Dynabeads antibody coupling kit were purchased from Invitrogen (Carlsbad, CA, USA). NanoDrop Lite Spectrphotometer, Bolt 4–12% Bis-Tris Plus Gels, and Amicon Ultra-15, 10K centrifugal filter were purchased from Thermo Fisher Scientific (Waltham, MA, USA).

All SPR assays were performed using Reichert Duel Channel SR7500DC SPR System and its associated software called Integrated SPRAutolink Version 1.1.14-T (Reichert Technologies, Buffalo, NY, USA). TraceDrawer Version 1.6.1 by Ridgeview Instruments AB (Upsala, Sweden) was used to process and analyze the SPR data. A 500 kDa carboxymethyl dextran hydrogel surface sensor chip (SR7000 gold sensor slide) was purchased from Reichert Inc., NY, USA. Monoclonal antibodies (MAb 1E10 and MAb 1C8) specific to *S.* Typhimurium flagellin were produced in our laboratories as described in a previous study [29].

### 2.2. Preparation of SPR Sensor Surface

The 500 kDa carboxymethyl dextran hydrogel surface sensor chip (SR7000 gold sensor slide) was installed onto a Reichert SR7500DC biosensor following the manufacturer’s instruction. The sensor surface was then pre-conditioned by running PBST at 20 µL/min until a stable baseline was obtained. The flow rate of 20 µL/min and temperature of 25 °C were maintained throughout the immobilization process. Immobilization of MAb 1E10 on the sensor surface was performed as descried in our previous study [29]. In order to activate carboxyl groups on the surface of the sensor chip, fresh preparation of 40 mg EDC and 10 mg NHS dissolved in 1 mL of water was injected onto the sensor surface for 5 min. After surface activation, MAb 1E10 diluted in 10 mM sodium acetate (150 µg/mL, pH 5.2) was injected to the left channel of the surface for 5 min. Then, BSA dissolved in 10 mM sodium acetate (75 µg/mL, pH 5.2) was injected to both channels to saturate the remaining active sites. Finally, quenching solution (1.0 M ethanolamine, pH 8.5) was injected for 5 min to deactivate the carboxyl groups, and to wash away the unbound antibody and BSA. A continuous flow of running buffer (PBST) at 20 µL/min was maintained after the completion of antibody immobilization. SPR assays were carried out after a stable baseline was achieved. All of the experiments were performed at a constant temperature of 25 °C. Filtered and degassed 1X PBST was used as the running buffer.

### 2.3. SPR Assay Formats

Three assay formats were designed utilizing single or paired monoclonal antibodies (MAb 1E10 and MAb 1C8) specific to the flagellin of *S.* Typhimurium. The first was a direct assay which employed MAb 1E10 immobilized on the sensor surface to capture the flagellin (Figure 1a). Sample preparation was directly injected without further processing and SPR response was directly related to the flagellin captured by the MAb 1E10 on the sensor surface. The second was a sequential two-step sandwich assay which applied the injection of MAb 1C8 (24 µg/mL) after the direct assay to form a sandwich on the captured flagellin (Figure 1b). The MAb 1C8 provides further amplification to the SPR response of the captured flagellin in the direct assay. In this format, two responses were recorded one from the direct assay and another from the binding of MAb 1C8 to the captured flagellin. The third was a pre-incubation one-step sandwich assay which applied incubation (30 min, room temperature) of MAb 1C8 (24 µg/mL) with sample preparation before injection (Figure 1c). The flagellin-MAb 1C8 complex formed during the incubation contributed to an augmented SPR response than the flagellin alone when captured by MAb 1E10 on the sensor surface.

### 2.4. Preparation of S. Typhimurium Flagellin

Cultures of *S.* Typhimurium were prepared on TSA plates and incubated at 35 °C for 24 h after retrieving from the freezer. The cultures were further propagated by transferring to additional TSA plates. Cells from each plate after incubating at 35 °C for 24 h were collected by washing with 1 mL of PBS. The recovered suspensions were centrifuged (3000× *g*, 10 min) and the cell pellets were collected. Flagellin proteins were extracted by adding a volume of 10 mL of glycine-HCl (250 mM, pH 2.0) to the pellets. After mixing and holding the suspension for 30 min at room temperature, centrifugation (16,500× *g*, 10 min) was done to collect the supernatant. The pH of the supernatant was adjusted to 7.0, and an Amicon Ultra-15, 10 K centrifugal filter was used to concentrate and to exchange buffer to PBS. Finally, the retained liquid in the filter was collected and the volume was adjusted to 500 μL using PBS. Protein concentration was measured using a NanoDrop Lite spectrophotometer. The flagellin preparations were analyzed using Bolt 4–12% Bis-Tris Plus gels to check for purity. Presence of minor fragments in the flagellin preparation was noticed as described in our previous study [29]. The flagellin preparations were stored at −80 °C.

### 2.5. Preparation of S. Typhimurium Contaminated Romaine Lettuce

*S.* Typhimurium was cultured on TSA plates and colonies were picked and suspended in 5 mL of PBS. The suspension was further diluted to yield a serial of 10-fold dilutions. Romaine lettuce samples (25 g each) were inoculated by pipetting 100 µL of the suspension of *S.* Typhimurium onto the surface of the samples which were then incubated in 225 mL of BPW or LB for 24 h at 35 °C. The numbers of aerobic bacteria and *S.* Typhimurium in the diluted suspensions and the enriched BPW and LB samples were determined by the plate count method using plate count agar and XLT-4 selective agar, respectively. The commensal bacteria that grow in BPW enriched romaine lettuce samples were selected on their different morphologies from the TSA plates. These colonies were isolated and passed on TSA plates several times to acquire pure cultures. Each of these cultures was further tested by API-20 E, a standardized identification system which uses 21 miniaturized biochemical tests and a database for identification of Enterobacteriaceae and other non-fastidious, Gram negative rods.

### 2.6. Analytical Procedures of SPR Assay

The contaminated romaine lettuce sample (25 g) was incubated in 225 mL of BPW or LB for 24 h at 35 °C. A volume of 1 mL was taken from the enriched culture medium of the contaminated romaine lettuce and centrifuged at 16,500× *g* for 5 min. The pellet of bacteria cells was collected and suspended in 100 µL of glycine-HCl (250 mM, pH 2.0) to extract the flagellin proteins. After holding the suspension for 30 min at room temperature, centrifugation was done to collect the supernatant. A volume of 200 µL of 1.5X PBST was mixed with the supernatant and this sample preparation is used for the injection in the SPR assay. A continuous flow of running buffer (PBST) at 20 µL/min was maintained. SPR assays were carried out after a stable baseline was achieved. All of the experiments were performed at a constant temperature of 25 °C.

### 2.7. Flagellin Purification Using Immunomagnetic Separation

In one of the experiments (Section 3.5), the sample preparation from above (Section 2.6) was further purified by the immunomagnetic separation. Dynabeads were used to capture and separate flagellin before the SPR assay. MAb 1C8 was coupled to Dynabeads (35.5 µg/mg) following the manufacture’s instruction. Aliquots of each bead suspension captured by Dynal magnet were washed three times with 1X PBST. Sample preparations were incubated with 250 µL of the MAb 1C8-coupled Dynabeads (10 mg/mL) for 30 min at room temperature with constant shaking using a Hula mixer. After incubation, the beads were separated from the sample preparation using a Dynal magnet and washed three times with 500 µL of 1X PBST. Flagellin bound to the beads was then released by incubating with 125 µL of glycine-HCl (250 mM, pH 2.0) under constant shaking for 5 min using a Hula mixer. Finally, the supernatant was collected and diluted with 375 µL of 1.2X PBST. This purified flagellin preparation was then used in the SPR assay.

### 2.8. Data Analysis

All experiments were conducted in replicated samples and repeated SPR runs. Averages and standard deviations of data from SPR assays were calculated. Analysis of variance (ANOVA) and multiple comparisons were performed with the IBM SPSS Statistics 24 software (Armonk, NY, USA). Tukey’s post hoc tests were performed for multiple comparisons.

## 3. Results

### 3.1. SPR Sensorgram and Dose Response Curve

Serial dilutions of flagellin preparations (9.6, 4.8, 2.4, 1.2, 0.6, and 0 µg/mL) of *S.* Typhimurium were analyzed by the direct assay. The SPR sensorgrams showing the association and dissociation of flagellin at various concentrations captured by the anti-flagellin antibody (MAb 1E10) immobilized on the sensor surface are shown in Figure 2a. A dose response curve was constructed to demonstrate the relation of the flagellin concentrations and the SPR responses at the peak of the association phase. A linear relation between the SPR responses and flagellin concentrations was observed within the experimental range (Figure 2b). The detection limit of the direct assay was 0.2 µg/mL. Concentration dependency of SPR responses indicated necessity of an efficient extraction protocol to recover *S.* Typhimurium flagellin from the complex food matrixes.

### 3.2. SPR Assay of Contaminated Romaine Lettuce Using Two Enrichment Methods

Samples of romaine lettuce were inoculated with *S.* Typhimurium at 4.1 log cfu/g and incubated in one of the two enrichment medias (BPW or LB for 24 h at 35 °C). The direct assay was applied to detect *S.* Typhimurium in romaine lettuce after enrichment and the results are presented in Figure 3. The average concentration of *S.* Typhimurium in the enriched BPW medium (8.6 log cfu/mL) was higher than that of the enriched LB medium (7.3 log cfu/mL). Similarly, the SPR responses of the direct assay on two enriched media showed significant differences. The average SPR response of BPW enriched samples (69.4 µRIU) was significantly higher than that of the LB enriched samples (44.3 µRIU). The pre-enrichment step provides two important functions, first to achieve the level of *S.* Typhimurium detectable by the SPR assay, and second to assure the presence of viable *S.* Typhimurium in the samples. It is important for any molecular based (DNA or protein) assay to confirm that the detection signal is not a false positive from the non-viable cells. In our observation, the two enrichment protocols yielded significant difference in the *S.* Typhimurium concentrations after overnight incubation and subsequently affect the SPR responses but both enrichment methods were sufficient to detect any positive samples. There are other media that can be used for the enrichment, when different types of food samples are analyzed. Overall, the results showed that BPW is a better enrichment medium than LB for *S.* Typhimurium in romaine lettuce, and provides better SPR responses as compared to LB. Therefore, BPW was used for the enrichment of *S.* Typhimurium in the following experiments.

### 3.3. SPR Assay of Romaine Lettuce with Different Contamination Levels

The capability of the SPR assay to detect *S.* Typhimurium at a low level of contamination is crucial for the successful application as a food safety surveillance tool. We have demonstrated the ability of the SPR assay to detect *S.* Typhimurium at such low contamination levels. The direct assays were applied to detect *S.* Typhimurium in romaine lettuce samples of different contamination levels using BPW enrichment. Irrespective of the original contamination levels, the concentration of *S.* Typhimurium of all samples increased to more than 8 log cfu/mL after the enrichment. As shown in Figure 4, samples contaminated with *S.* Typhimurium at 0.9, 1.9, 2.9, and 5.9 log cfu/g after overnight enrichment resulted in 8.3, 8.7, 8.8, and 9.0 log cfu/mL in the BPW, respectively. It was noted that samples with lower contamination level (0.9 log cfu/g) yielded a slightly lower concentrations of *S.* Typhimurium (8.3 log cfu/mL) after enrichment than those with higher contamination levels (1.9, 2.9, and 5.9 log cfu/g). Nevertheless, the contamination of *S.* Typhimurium at such a low level (0.9 log cfu/g) can be enriched overnight in BPW to a concentration which can be easily detected by the direct assay.

### 3.4. SPR Assay of Commensal Bacteria Isolated from Romaine Lettuce

Romaine lettuce samples enriched in BPW showed considerable numbers of commensal bacteria that generated colonies of different morphologies in addition to *S.* Typhimurium. These colonies were isolated and cultured individually on TSA plates to obtain pure cultures of each. Seven bacteria, including *Enterobacter cloacae*, *Pseudomonas fluorescens*, *Aeromonas salmonicida*, *Serratia* spp., *Brucella* spp., *Photobacterium damselae*, and *Pseudomonas aeruginosa* were isolated and presumptively identified. All of them were separately grown in TSA agar to obtain a single culture. These commensal bacteria and *S.* Typhimurium were then cultured individually in BPW (24 h at 35 °C) and the concentrations of bacteria in BPW reached a range between 9 and 10 log cfu/mL after overnight incubation. Direct assay was performed using equal concentrations of bacteria (9.0 log cfu/mL) diluted from each of the enriched pure cultures and the SPR responses of these commensal bacteria are presented in Figure 5. *S.* Typhimurium generated significantly higher SPR response; whereas all other seven isolated bacteria generated SPR responses close to the baseline. The average of the SPR response for *S.* Typhimurium was 97.1 µRIU, and for other bacteria was less than 2.5 µRIU. The results indicated a higher specificity of the SPR assay for the detection of *S.* Typhimurium in the presence of other commensal bacteria.

### 3.5. SPR Assay of Contaminated Romaine Lettuce Using Immunomagnetic Separation

Romaine lettuce samples inoculated with different levels of *S.* Typhimurium were pre-enriched in BPW. Enriched samples were individually prepared and flagellin in the sample preparation was further purified using magnetic Dynabeads covalently coupled with MAb 1C8 (Section 2.7). The functionalized magnetic beads captured flagellin and separated it from the sample preparation in a higher purity. The crude sample preparation and purified flagellin were both evaluated by the direct assay. The results indicated that the direct injection of sample preparation yielded significantly higher (*p* < 0.05) SPR response than that of purified flagellin. The SPR responses from the direct injection of sample preparations produced more than 57.6 µRIU for romaine lettuce samples contaminated with various levels of *S.* Typhimurium (Figure 6). In contrast, the SPR responses from injection of purified flagellin samples were significantly lower (all less than 17.6 µRIU). Immunomagnetic separation is a laboratory tool that can efficiently isolate proteins of interest from a complex matrix. However, the results suggested that further purification of flagellin from the sample preparation did not improve the detection sensitivity of the SPR assay. It appears that presence of other proteins in the sample preparation did not interfere with the assay. The quantity of flagellin which was partially lost during the immunomagnetic separation process accounted for the reduced SPR response. Much large quantity of the beads will be required to increase the concentration yield but this approach will not be practical because more antibodies will be needed in the analytical procedure. Therefore, we suggest that immunomagnetic separation is not a necessity to improve the sensitivity of the SPR assay.

### 3.6. SPR Assay of Contaminated Romaine Lettuce Using Different Assay Formats

Three SPR assay formats, direct assay, sequential two-step sandwich assay and pre-incubation one-step sandwich assay, as illustrated in Figure 1, were applied to detect *S.* Typhimurium at various concentrations and the performances of three assay formats were evaluated. Romaine lettuce samples contaminated with *S.* Typhimurium were incubated in BPW (24 h at 35 °C). Numbers of *S.* Typhimurium increased to 1.9 × 10^8^ cfu/mL (8.3 log cfu/mL) after enrichment. Enriched samples were serially diluted in fresh BPW to produce various concentrations of *S.* Typhimurium between 4.7 × 10^5^ and 9.5 × 10^6^ cfu/mL (5.7 and 7.0 log cfu/mL), which were analyzed by the three different assay formats. The results indicated that pre-incubation one-step sandwich assay generated highest SPR response than that of the direct assay, and sequential two-step sandwich assay (Figure 7).

The SPR responses of the sequential two-step sandwich assay increased by 1.6 times as compared to that of the direct assay. The molecular mass of MAb 1C8 (150 kDa) is about three times larger than flagellin (52 kDa). However, the SPR responses generated from the sequential injection of MAb 1C8 did not reach that level because the concentration of MAb 1C8 (24 µg/mL) was not sufficient to saturate all of the captured flagellin on the sensor surface. In contrast, the SPR responses of pre-incubation one-step sandwich assay increased by 3.2 times as compared to that of the direct assay. The molecular mass of flagellin-MAb 1C8 complex (202 kDa) formed during the pre-incubation was about four times larger than the flagellin alone. Therefore, pre-incubation will be a more efficient way of amplifying SPR response when the same concentration of MAb 1C8 (24 µg/mL) is used in the experiments.

The lowest concentrations of *S.* Typhimurium in the enriched samples that can be detected by the direct assay and sequential two-step sandwich assay were 1.9 × 10^6^ and 1.6 × 10^6^ cfu/mL (6.3 and 6.2 log cfu/mL), respectively. The lowest concentration of *S.* Typhimurium in the enriched samples that can be detected by the pre-incubation one-step sandwich assay was 4.7 × 10^5^ cfu/mL (5.7 log cfu/mL). Although all three assay formats can easily detect a low level of contamination after enrichment, the pre-incubation one-step sandwich assay is recommended for the enhanced SPR response and the increased assay sensitivity.

## 4. Discussion

Pre-enrichment is a prerequisite for the detection of *Salmonella* in culture methods and in molecule methods such as real-time PCR. BPW and LB are the common pre-enrichment media included in most of the standard methods [5,30]. For the enrichment of *S.* Typhimurium, various studies have successfully used BPW [31,32,33,34] and LB [35,36,37] in oysters, shrimps, chicken, eggs, animal feed, sprout, cilantro, cantaloupes, carrot, cucumber, and pre-packed mixed-salad. Because the growth profiles of *Salmonella* can be affected by the types of foods [38], in this study, we have compared BPW and LB for their enrichment potential of *S.* Typhimurium in romaine lettuce. We have showed that BPW can enrich *S.* Typhimurium in romaine lettuce better than LB and a contamination level as low as 0.9 log cfu/g in romaine lettuce can be increased to more than 8 log cfu/mL overnight using BPW. This finding is in accordance with those of the previous investigations in which BPW multiplied *Salmonella* serovars including *S.* Typhimurium to more than 6 log cfu/mL in sprouts and other foods [38].

Immunomagnetic separation is a popular alternative for the purification of proteins from the matrixes of biological molecules, without requiring column separations and centrifugation steps [39]. Various studies have utilized immunomagnetic separation to isolate *Salmonella* cells [40,41,42,43] for the detection purposes in conjunction with the culture methods or molecular methods. However, there has not been a report of using immunomagnetic separation to capture and purify *S.* Typhimurium flagellin for the detection purpose. In our study, we isolated flagellin from the sample preparations using monoclonal antibody (MAb 1C8) coated magnetic beads in a belief that the purified flagellin would give higher SPR responses as compared to the crude sample preparations. However, the results showed that the immunomagnetic separation purified flagellin reduced the SPR responses in the direct assay. This can attribute to the facts that the immunomagnetic separation is not able to recuperate all the flagellin in the sample preparation due to the limited concentration of the MAb 1C8 coated magnetic beads.

We have evaluated three SPR assay formats using a set of romaine lettuce samples with various contamination levels of *S.* Typhimurium. As compared to the direct assay, the sequential two-step sandwich assay and pre-incubation one-step sandwich assay have significantly increased the SPR responses by an average of 60% and 220%, respectively. This increment is due to the higher molecular weight of MAb 1C8 (150 kDa) in the sequential two-step sandwich assay and the flagellin (52 kDa)-MAb 1C8 (150 kDa) complex (202 kDa) formed during the incubation in the pre-incubation one-step sandwich assay [44] as compared to the flagellin alone (52 kDa) [45]. Our result is well explained by the fact that higher the molecular mass that binds to the bio-recognition molecule on the sensor surface, higher the change in refractive index and higher the SPR response [46]. Additionally, pre-incubation one-step sandwich assay is more sensitive than the sequential two-step sandwich assay when the equivalent concentrations of MAb 1C8 are used. In this study, we have reported that as low as 5.7 log cfu of *S.* Typhimurium can be detected using the SPR assays. This is in close agreement with the studies that were conducted in milk matrix [27] and in buffer system [23]. Another study has reported a detection sensitivity of 2 log cfu, which can be due to the improvement of immobilized antibody orientation using protein G and the performance of experiments in a buffer system without the involvement of food matrix [25]. Nevertheless, we have presented that as low as 0.9 log cfu/g of *S.* Typhimurium in romaine lettuce can be detected following the enrichment in BPW. The sensitivity of the SPR assay can be further improved by using antibody-coupled nanoparticles [47]. High molecular mass of antibody-coupled nanoparticles has potential to further amplify the detection signal [46]. Alternatively, concentration devices such as vacuum filtration with filter membrane [48] can increase the recovery of *S.* Typhimurium from the samples.

Monoclonal antibodies used in this study are specific to the flagellin (phase 1 and phase 2) of *S.* Typhimurium, and does not recognize ligands other than flagellin [49]. Additionally, these monoclonal antibodies are well characterized and recognize distinct epitopes on the flagellin of *S.* Typhimurium [29]. The specificity of the SPR assay using these monoclonal antibodies has been further confirmed with seven commensal bacteria isolated from the romaine lettuce samples, including *Enterobacter cloacae*, *Pseudomonas fluorescens*, *Aeromonas salmonicida*, *Serratia* spp., *Brucella* spp., *Photobacterium damselaeF* and *Pseudomonas aeruginosa*. Some of these are opportunistic human pathogens such as *Pseudomonas aeruginosa*, *Brucella* spp., and *Serratia* spp.; while some are animal pathogens, such as *Photobacterium damselae* and *Aeromonas salmonicida.* It is important to note that fresh vegetables can harbor large and diverse populations of bacteria. Therefore, the specificity of a SPR assay is even far more important than the sensitivity and should be demonstrated when validating the SPR assay.

## 5. Conclusions

We have demonstrated a label-free SPR detection method for *S.* Typhimurium using flagellin specific monoclonal antibodies. By comparing different SPR assay formats, we have also demonstrated that the pre-incubation one-step sandwich assay provides an enhanced signal and has the ability to detect lower number of *Salmonella* than the direct assay and sequential two-step sandwich assay. In addition, we have showed that low levels of *S.* Typhimurium contamination in leafy vegetables can be successfully detected following the BPW enrichment. SPR assays for other serotypes of *Salmonella* can be developed on the same ground of this work using serotype-specific antibodies. This study also suggests the possibility of exploring the use of nanoparticles in the SPR assay to further improve the detection sensitivity.

## Figures and Tables

**Figure 1 biosensors-09-00094-f001:**
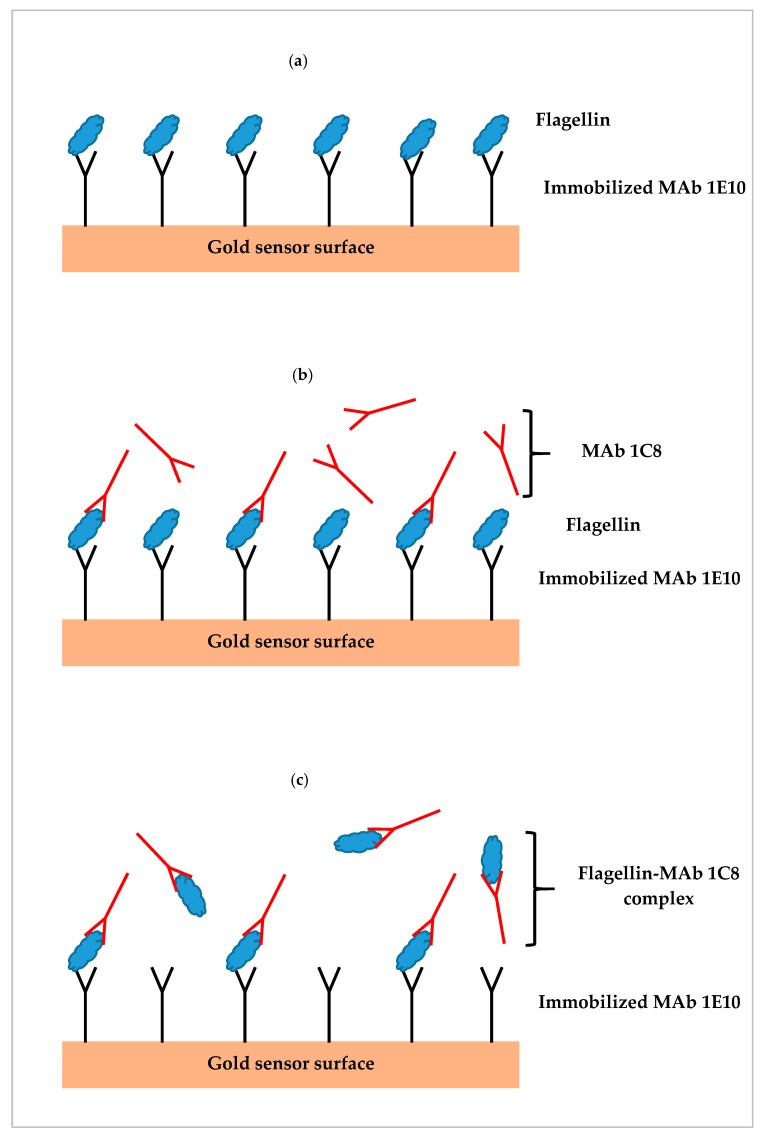
Illustrations of three different surface plasmon resonance (SPR) assay formats; (**a**) direct assay—sample preparation is injected directly, (**b**) sequential two-step sandwich assay—MAb 1C8 is injected following sample injection, and (**c**) pre-incubation one-step sandwich assay—sample preparation is pre-incubated with MAb 1C8 before injection.

**Figure 2 biosensors-09-00094-f002:**
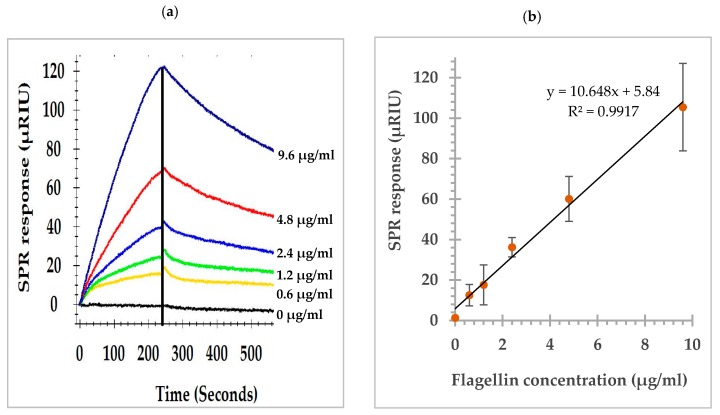
(**a**) SPR sensorgrams of the direct assay showing the association and dissociation of flagellin at various concentrations with the anti-flagellin antibody (MAb 1E10) immobilized on the sensor surface; (**b**) SPR responses at the peak of association phase, the vertical line in (**a**), showing a linear relation to the flagellin concentrations.

**Figure 3 biosensors-09-00094-f003:**
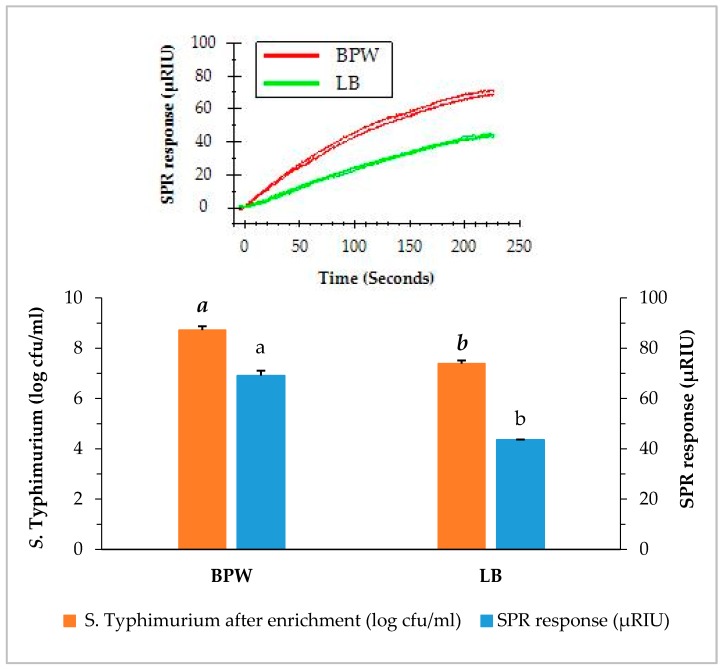
Concentrations of *S.* Typhimurium in romaine lettuce samples after enrichment and the SPR responses when the enriched samples were analyzed by the direct assay. Romaine lettuce samples contaminated with *S*. Typhimurium were incubated individually in buffered peptone water (BPW) and lactose broth (LB) at 35 °C for 24 h. The significant differences (*p* < 0.05) were indicated by different letters.

**Figure 4 biosensors-09-00094-f004:**
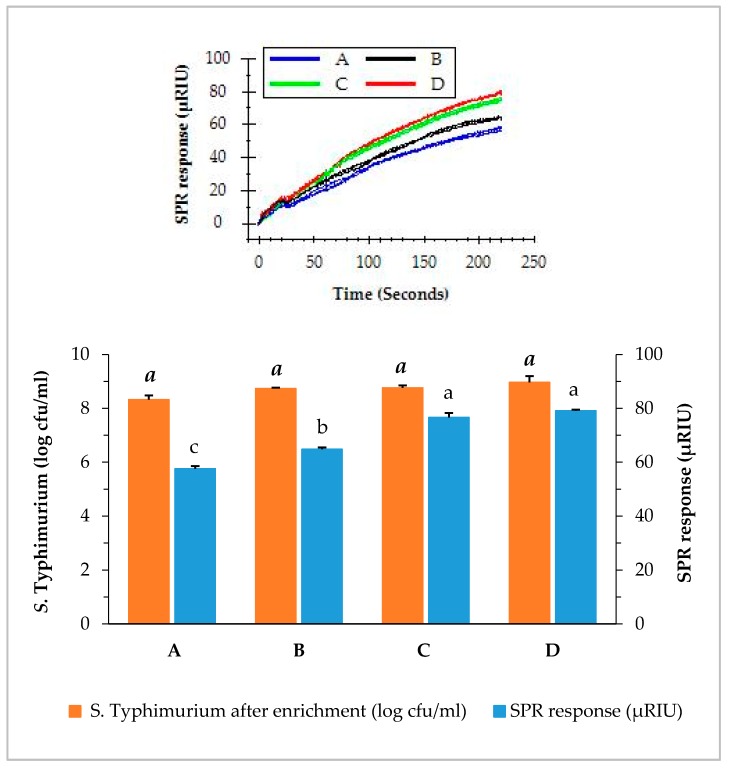
Concentrations of *S.* Typhimurium in romaine lettuce samples after enrichemnt and SPR responses when the enriched samples were analyzed by the direct assay. Romaine lettuce samples (**A**–**D**) contaminated with various levels of *S.* Typhimurium (0.9, 1.9, 2.9, and 5.9 log cfu/g) were incubated in BPW at 35 °C for 24 h. The significant differences (*p* < 0.05) were indicated by different letters.

**Figure 5 biosensors-09-00094-f005:**
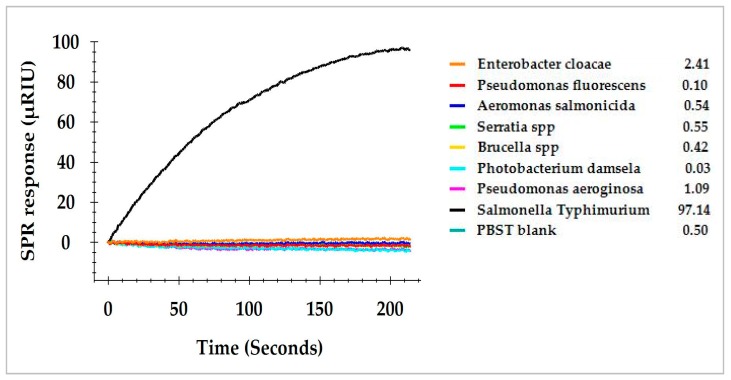
SPR responses of *S*. Typhimurium and seven commensal bacteria isolated from the enriched romaine lettuce samples and analyzed by the direct assay. Commensal bacteria in the enriched samples were isolated on TSA plates and individual colonies were identified and cultured in BPW.

**Figure 6 biosensors-09-00094-f006:**
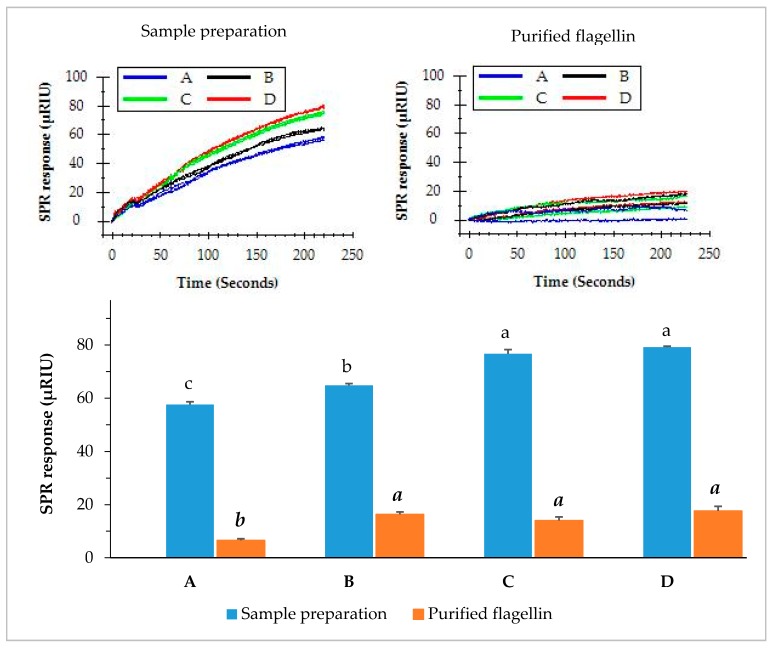
SPR responses of the romaine lettuce samples after enrichment when analyzed by the direct assay. Romaine lettuce samples (**A**–**D**) contaminated with various levels of *S.* Typhimurium (0.9, 1.9, 2.9, and 5.9 log cfu/g) were incubated in BPW at 35 °C for 24 h. Sample preparations were either directly used for the assay or further purified with immunomagnetic separation before the assay. The significant differences (*p* < 0.05) were indicated by different letters.

**Figure 7 biosensors-09-00094-f007:**
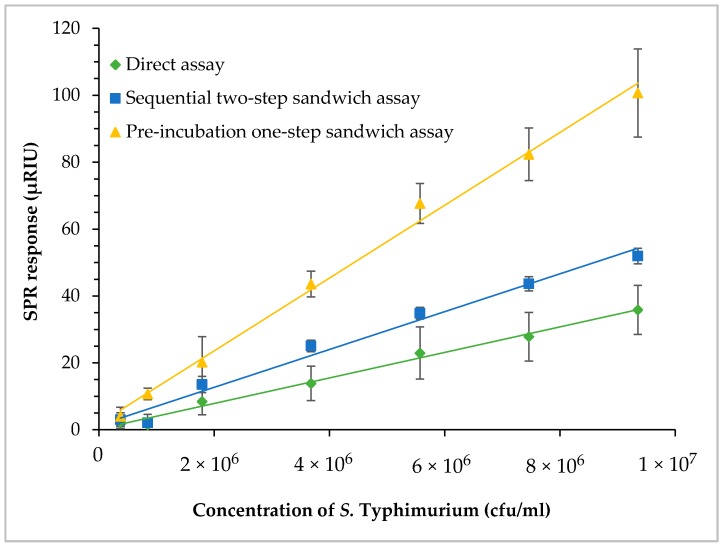
Comparisons of the three assay formats—direct assay, sequential two-step sandwich assay, and pre-incubation one-step sandwich assay. Romaine lettuce samples contaminated with *S.* Typhimurium were incubated in BPW at 35 °C for 24 h. The enriched samples were further diluted to various concentrations of *S.* Typhmurium before the assays.

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
