# Peer review of "Detection of Salmonella Typhimurium in Romaine Lettuce Using a Surface Plasmon Resonance Biosensor"

_biosensors, 2019, doi:10.3390/bios9030094_

Reviewer 1 Report

The reviewed paper concerns the use of the SPR technique for determination of the flagellin of Salmonella enterica serovar Typhimurium and for determination of the number of colonies of the S. Typhimurium bacteria in Romaine lettuce. This is an excellent paper in terms of the tools used and its practical significance, and deserves to be published on a priority basis. Three highly specific SPR immunobiosensors for flagellin determination have been developed, as well as a procedure for conversion of the SPR signal for cfu of S. Typhimurium on leaves of Romaine lettuce. However, I recommend adding to the manuscript a precise analytical procedure beginning with lettuce leaves which contains instructions for the calibration of cfu of S. Typhimurium. Additionally, an explanation of how the flagellin protein is released by S.  Typhimurium deserves to be added. Therefore, minor revision is recommended.

Reviewer 2 Report

The paper by Bhandari et. al on the detection of Salmonella in romaine lettuce by SPR is very interesting and also is a good read to the audience of this journal. I have a few comments 

Please include a last line in the abstract how this study is helpful.

detection of flagellin by electrophoresis is not very well studied. Please have a standard flagellin run along with the prepared protein. 

please check for abbreviation wherever they were mentioned first in the text full form is required. 

please conduct a thorough spell check for spell mistakes.

mention the mol.wt of flagellin 

Injection of BSA to saturate binding sites might bring in non-specific binding and suppress actual binding. what was the pH of sod. acetate for BSA

please add inset of raw spr data of fig 3 and fig 4

"sample preparation did not improve the detection sensitivity of the SPR assay. It appears that the quantity of flagellin, which is partially lost in the immunomagnetic separation process, deems more important than the purity in the SPR assay." what does this mean? If immunomagnetic separation didn't yield satisfactory results, please change the protocol or standardize it.

raw data for fig 6

significance of fig 7, what method is best and recommended by you if you were to suggest a regulatory guidelines

  please explain " Contrary to our general belief, the results showed that the immunomagnetic 343 separations purified flagellin reduced the SPR responses. This could attribute to the facts that the 344 immunomagnetic separation is not able to recuperate all the flagellin in the sample preparation and 345 to preclude the loss of flagellin during washing steps." if immune separation failed then how do you know that the antibody binds to flagellin and not some nonspecific binding detected in the SPR.

what was the PBST, T mean tween 80 or 20 what is the concentration

please also comment on the LOD and detection limits set by regulatory agencies for the presence on the salmonella on the lettuce?

please explain amplified signal means here? what were the responses before amplification and why was an amplification necessary as flagellin is a big protein 60,000 Da.

Author Response

Round  2

Reviewer 2 Report

All queries are answered to satisfaction.